# Water Metabolism of *Lonicera japonica* and *Parthenocissus quinquefolia* in Response to Heterogeneous Simulated Rock Outcrop Habitats

**DOI:** 10.3390/plants12122279

**Published:** 2023-06-12

**Authors:** Xiaopan Zhao, Yanyou Wu, Deke Xing, Haitao Li, Furong Zhang

**Affiliations:** 1School of Agricultural Engineering, Jiangsu University, Zhenjiang 212013, China; 2State Key Laboratory of Environmental Geochemistry, Institute of Geochemistry, Chinese Academy of Sciences, Guiyang 550081, China; 3Department of Agricultural Engineering, Guizhou Vocational College of Agriculture, Qingzhen 551400, China

**Keywords:** bicarbonate stress, drought tolerance, *Lonicera japonica*, *Parthenocissus quinquefolia*, rock outcrop habitats, water metabolism

## Abstract

The karst carbon sink caused by rock outcrops results in enrichment of the bicarbonate in soil, affecting the physiological process of plants in an all-round way. Water is the basis of plant growth and metabolic activities. In heterogeneous rock outcrop habitats, the impact of bicarbonate enrichment on the intracellular water metabolism of plant leaf is still unclear, which needs to be revealed. In this paper, the *Lonicera japonica* and *Parthenocissus quinquefolia* plants were selected as experimental materials, and electrophysiological indices were used to study their water holding, transfer and use efficiency under three simulated rock outcrop habitats, i.e., rock/soil ratio as 1, 1/4 and 0. By synchronously determining and analyzing the leaf water content, photosynthetic and chlorophyll fluorescence parameters, the response characteristics of water metabolism within leaf cells to the heterogeneous rock outcrop habitats were revealed. The results showed that the soil bicarbonate content in rock outcrop habitats increased with increasing rock/soil ratio. Under the treatment of a higher concentration of bicarbonate, the leaf intra- and intercellular water acquisition and transfer efficiency as well as the photosynthetic utilization capacity of *P. quinquefolia* decreased, the leaf water content was lower, and those plants had low bicarbonate utilization efficiency, which greatly weakened their drought resistance. However, the *Lonicera japonica* had a high bicarbonate use capacity when facing the enrichment of bicarbonate within cells, the above-mentioned capacity could significantly improve the water status of the leaves, and the water content and intracellular water-holding capacity of plant leaves in large rock outcrop habitats were significantly better than in non-rock outcrop habitats. In addition, the higher intracellular water-holding capacity was likely to maintain the stability of the intra- and intercellular water environment, thus ensuring the full development of its photosynthetic metabolic capacity, and the stable intracellular water-use efficiency also made itself more vigorous under karstic drought. Taken together, the results suggested that the water metabolic traits of *Lonicera japonica* made it more adaptable to karst environments.

## 1. Introduction

As the most significant phenological feature in karst areas, rock outcrops (ROs) have important hydrological significance in karst ecosystems [1]. In habitats with outcrops, exposed rocks not only interrupt the mid-loam flow but also intercept a large proportion of rainwater transport to soil patches, thus making the moisture and resource content of soils around outcrops significantly higher than in habitats far from outcrops or without outcrops [2,3,4]. Carbonate rocks are the most dominant rock type in karst areas [5], and the products of their dissolution and decomposition under hydraulic action produce enrichment in adjacent soils with the flow of water, so that the soil bicarbonate (HCO_3_^−^) content is significantly higher than that in non-karst areas, which is the so-called carbon sink effect [6], and the concentration of HCO_3_^−^ in such a soil solution shows a positive correlation with soil moisture [7]. Moreover, the heterogeneous redistribution of precipitation, which is attributed to the various morphologies, sizes and orientations of ROs in habitats, makes the soil moisture content in the karst landscape area vary significantly, resulting in heterogeneous karst habitats with differential enrichment of bicarbonate [8], which further affects biodiversity distribution. The karst ecological environment is extremely fragile, and karstic desertification is of great concern today. In order to accelerate the process of ecological restoration in karst areas, the vulnerable ecological environment in these areas can only be better managed by selecting plants that are higher matched to heterogeneous karst habitats.

HCO_3_^−^ is an alkaline salt with strong stability in the environment. In HCO_3_^−^ habitats, the physiological activities in plants are highly susceptible to disturbance [9,10,11,12]. On the one hand, under the effects of a high concentration of HCO_3_^−^, the growth and function of plant roots are inhibited, and the excessive accumulation of HCO_3_^−^ in plants can also have toxic effects on plant cells, blocking cell metabolism [13]; on the other hand, the high pH induced by a high concentration of HCO_3_^−^ will change the ionic environment of the soil around plant roots, and the precipitation of ions such as Fe^2+^ and Mg^2+^ in soil habitats will lead to the obstruction of plant photosynthetic pigment synthesis, further limiting plant growth and metabolic processes [14,15]. However, it is worth mentioning that HCO_3_^−^ also has a positive effect on plant photosynthesis that cannot be ignored. HCO_3_^−^ not only acts as a mobile proton acceptor during the photosynthesis reaction to improve photosynthetic efficiency [16,17] but it also promotes photosynthesis in plants by enabling the reversible conversion of HCO_3_^−^ to water and carbon dioxide, under the efficient catalysis of carbonic anhydrase (CA) [18,19,20]. When plants are under stress conditions that trigger stomatal closure, this process can directly increase the assimilation of CO_2_, which is beneficial to plant growth and development.

Water is the basis of all plant growth and metabolic activities [21]. In recent years, a series of achievements has been made in the study of the physiological characteristics of plants under HCO_3_^−^ stress, but the related research has rarely included the changes in the water metabolism characteristics of plants, and the influence mechanism of HCO_3_^−^ on plant water metabolism remains unclear. In addition, the typical aboveground–underground dual spatial structure of karst has a very strong water seepage effect, so the temporary soil drought may occur even during periods of abundant rainfall, affecting the water metabolic processes of plants [22,23]. Therefore, the process of HCO_3_^−^ effects on plant water metabolism is often accompanied by drought in karst areas, which also poses further challenges to plant suitability studies.

Plant photosynthesis is very sensitive to changes in water availability, and studies have shown that when plants do not have access to sufficient water, stomatal conductance of plant leaves decreases, transpiration rate decreases, water-use efficiency increases and net photosynthetic rate decreases [24,25]. However, these photosynthetic parameters reflect the variation of intercellular water in leaves and do not characterize the internal water metabolism well, while the intracellular water metabolism of leaves is more reflective of plant response to karst drought. Plant electrophysiological techniques have an unmistakable advantage in reflecting the intracellular water metabolism of plant leaves [26,27]. In 1967, Sinyukhin’s work confirmed that plant electrical signals are present at the beginning and throughout a plant’s life, and almost all life activities in a plant’s body involve charge separation, electron movement and proton and dielectric transport [28]. Plant electrophysiological technology regards plant membranous cells and organelles as containers that can conduct electricity based on electrical principles (Figure 1), the liquid environment inside and outside the membrane is equivalent to the bipolar plates of a capacitor, membrane lipids having resistive properties and intracellular ions, ionic groups and electric dipoles are electrolytes. When the plant’s internal growth conditions and the external environment change, the permeability of the plant cell membrane alters initially, resulting in a change in the electrolyte concentration inside and outside the cells, which leads to changes in the cell’s capacitance (C), resistance (R) and impedance (Z), and the water inside the cells also changes. Jamaludin et al. [29] found that the impedance showed a significant negative relationship with leaf water status, and Yu et al. [30] also found that the electrophysiological parameters C, R and Z of plant leaves could characterize the response of plant growth and development to soil moisture. Furthermore, parameters based on leaves’ intrinsic electrophysiological parameters derived from leaves’ intracellular water-holding capacity, water translocation rate and water-use efficiency have also shown unique value in reflecting the characteristics of plant leaf water metabolism [31]. Due to the close relationship between plant leaf cell volume and physiological C, R and Z, electrophysiological parameters can also be used to monitor plant water conditions online, quickly, accurately and in a timely and undamaged manner; thus, electrophysiological characteristics are increasingly applied to reflect plant water metabolism and diagnostic plant water status [32,33,34,35], which is a key consideration for the introduction of electrophysiological techniques in this study.

Considering the soil in karst areas is prone to HCO_3_^−^ enrichment and frequent drought, adapted plants often need to show efficient water metabolism under HCO_3_^−^ stress and drought stress simultaneously. Vines are pioneer plants in the succession of forest communities in rocky desertification areas [36], of which the *Lonicera japonica* (*L. japonica*) is highly adaptable, not strictly selective in terms of soil and climate and can be tolerant to drought, cold and salt in order to grow on limestone soils and generally saline lands, making it an excellent medicinal, silvicultural and greening plant [37,38]. The *Parthenocissus tricuspidate* (*P. tricuspidate*), an ideal plant for the greening of vine plants, is also highly adaptable to soil and climate [39]. It is resistant to cold, drought and barren, it has adaptability to soil acidity and alkalinity, and it is often used to restore vegetation on arid and barren hills. Both species show strong potential for karst habitability [40]. However, current studies on the water metabolism capacity of plants in karst areas have mostly considered the effects of a single drought stress, with little attention paid to the toxic effects of HCO_3_^−^, which is widespread in karst areas in addition to drought. The specific mechanisms of plant water metabolism in response to HCO_3_^−^ stress in karst habitats are also unclear.

Therefore, in this study, we analyzed the electrophysiological, photosynthetic and chlorophyll fluorescence parameters of *Lonicera japonica* and *P. quinquefolia* in heterogeneous simulated RO habitats under different drought levels in the field, aiming to (1) analyze the effects of heterogeneous ROs; (2) further explore the water metabolic characteristics of plants in heterogeneous RO habitats under drought conditions; (3) compare the matching of *L. japonica* and *P. quinquefolia* with karst heterogeneous RO habitats based on plant water metabolic characteristics. The results of the study provide a theoretical basis for the selection of plant species that match well with karst heterogeneous habitats in the process of vegetation restoration of fragile karst habitats.

## 2. Results

### 2.1. Differences in the Concentrations of HCO_3_^−^ in Three Simulated Habitats under Different Drought Periods

The results of HCO_3_^−^ content per 100 cm³ of soil in the simulated habitats with large rock outcrops in the field (L-ROs), small rock outcrops (S-ROs) and no rock outcrops (N-ROs) are shown in Figure 2. It was found that under the same drought period, the soil HCO_3_^−^ content in the simulated habitats showed a trend of L-RO habitat > S-RO habitat > N-RO habitat, which was consistent with the field reality.

### 2.2. Differential Analysis of Water Content and Electrophysiological Parameters of L. japonica and P. quinquefolia Leaves in Three Simulated Habitats under Different Drought Periods

As shown in Figure 3, under the same drought period, *L. japonica* showed difference between habitats only in the moderate drought period, with the LWC value of *L. japonica* in the L-RO habitat being 1.06 times higher than that in the N-RO habitat. *P. quinquefolia* showed significant difference between habitats in both the mild and moderate drought periods; for example, the LWC value of *P. quinquefolia* in the L-RO habitat was only 0.94 times higher than that in the N-RO habitat during the mild and moderate drought periods.

In the same simulated habitat, the LWC of *L. japonica* planted in S-RO and N-RO habitats decreased significantly with the development of the drought period compared to the previous drought period, while the LWC of *L. japonica* planted in the L-RO habitats was significantly lower in the moderate drought period than in the mild drought period, its value being 0.93 of that in the mild drought period. The LWC of *P. quinquefolia* in all three habitats decreased significantly with the development of the drought period.

As shown in Table 1, during the same drought period, the intracellular water holding capacity (IWHC) of *L. japonica* was enhanced with increasing rock-to-soil ratio in the simulated habitats, especially in the non- and moderate drought periods, this value was 1.88 in the L-RO habitat and 2.22 in the N-RO habitat; meanwhile, the IWHC of *P. quinquefolia* in the L-RO habitat was significantly lower than that in the N-RO habitat, at 0.49 times the IWHC of the N-RO habitat during non-drought, 0.47 times during mild drought and 0.51 times during moderate drought.

In the same simulated habitat, the IWHCs of *L. japonica* in the L-RO, S-RO and N-RO habitats in the moderate drought period were 52%, 59% and 45% of the non-drought period, and the IWHCs of *P. quinquefolia* were 38%, 31% and 37% of the non-drought period.

The change in leaf intracellular water transfer rate (IWTR) of *L. japonica* and *P. quinquefolia* among different conditions is presented in Table 2, and it appeared as a similar trend to that of IWHC shown in Table 1.

For the IWTR of *L. japonica* in the L-RO habitat, it was 1.96, 1.43 and 2.27 times that of the N-RO habitat during the non-, mild and moderate drought periods, respectively, while for the *P. quinquefolia* in the same habitat it was 0.47, 0.46 and 0.51 times, respectively.

During the same moderate drought, the IWTR of *L. japonica* in the L-RO, S-RO and N-RO habitats was 51%, 61% and 43% of that in the non-drought period, and for *P. quinquefolia* in the three habitats during moderate drought, the IWTR was 39%, 30% and 36% of that in the non-drought period.

In Table 3, under the same drought period, there was no difference in the IWUE of *L. japonica* between simulated habitats; the IWUE of *P. quinquefolia* increased with increasing habitat rock-to-soil ratio, but it was not significant.

With the development of drought, the increased IWUE value of *L. japonica* in three habitats was not significant, whereas the IWUE of *P. quinquefolia* in the L-RO habitat increased significantly by 58% during the stage of the mild to moderate drought periods.

### 2.3. Differential Analysis of Photosynthetic Parameters of L. japonica and P. quinquefolia Leaves in Three Simulated Habitats under Different Drought Periods

During the same drought period, the P_N_ showed a decreasing trend with increasing rock/soil ratio in simulated habitats (Figure 4). The P_N_ of both plants in the L-RO habitat was significantly lower than that in the respective N-RO habitat during the non-drought period. In mild and moderate drought periods, the P_N_ of *L. japonica* did not differ between habitats, and the P_N_ of *P. quinquefolia* in the L-RO habitat showed a significantly lower value than that in the N-RO habitat.

In the same simulated habitat, the P_N_ of both *L. japonica* and *P. quinquefolia* showed significant decreases with the development of drought degree, but the P_N_ of *L. japonica* was always higher than that of *P. quinquefolia*.

As Figure 5 shows, the stomatal conductance (gs) of *L. japonica* did not significantly differ between habitats during all drought periods; the gs of *P. quinquefolia* was also not significantly different between habitats in the non- and moderate drought periods, but during the mild drought period it was 0.024 in the L-RO habitat, which was significantly lower than that in the N-RO habitat with a value of 0.11.

With the development of drought, the gs of *L. japonica* and *P. quinquefolia* in all stimulated habitats decreased significantly, and during the period of moderate drought, the gs values of both plants were close to zero.

The change in instantaneous water-use efficiency (WUE_i_) is illustrated in Figure 6. There was no apparent difference in the WUE_i_ of *L. japonica* between habitats in each period; the WUE_i_ of *P. quinquefolia* in mild and moderate drought periods was significantly higher in the L-RO and S-RO habitats than in the N-RO habitat.

With the development of drought degree, the WUE_i_ of *L. japonica* and *P. quinquefolia* in the three habitats increased significantly. The WUE_i_ of *L. japonica* in the L-RO, S-RO and N-RO habitats increased by 119%, 141% and 112% from the non- to moderate drought periods, respectively; the WUE_i_ of *P. quinquefolia* increased by 270%, 275% and 146%, respectively.

### 2.4. Differential Analysis of Fluorescence Parameters of L. japonica and P. quinquefolia Leaves in Three Simulated Habitats under Different Drought Periods

Chlorophyll fluorescence induction kinetics reflects the intrinsic processes of plant photosynthesis, and by tracking changes in fluorescence parameters such as F_v_/F_m_, qP and ETR, they can further explore the photosynthetic structure and functional integrity of plants and refine their stress response characteristics.

The F_v_/F_m_ reflects the maximum light energy conversion efficiency within the PS II reaction center and is a good indicator of whether the photosynthetic machinery of the crop is impaired and the efficiency of light energy conversion [41].

As shown in Figure 7, during the same drought period, there was no difference in the F_v_/F_m_ of *L. japonica* between habitats, whereas the F_v_/F_m_ of *P. quinquefolia* in RO habitats was lower than that in the N-RO habitat and became more pronounced in the L-RO habitat.

With the development of drought, the value of F_v_/F_m_ of *L. japonica* in all habitats was consistently maintained at a high level.; however, *P. quinquefolia*’s F_v_/F_m_ in the same habitat decreased significantly.

The photochemical fluorescence quenching coefficient (qP) is a measure of the oxidation state of PSII primary electron acceptor QA, reflecting the efficiency of converting light energy captured by PSII antenna pigments into chemical energy [42]. The results in qP of the experimental plants are presented in Figure 8.

Among different habitats, the qP of *L. japonica* in the L-RO habitat was significantly lower than that in the S-RO habitat during the moderate drought, and no significant difference was observed between habitats during the other periods; the qP of *P. quinquefolia* in the L-RO habitat was significantly lower than that in the other habitats during the mild drought period and showed no apparent difference between habitats during the non- and moderate drought periods.

When the drought developed from non- to mild drought, the qP of *L. japonica* in the same habitat did not change significantly, while that of *P. quinquefolia* in the same habitat decreased significantly; when the drought further developed to moderate, the qP of both plants decreased significantly. Nevertheless, the qP of *L. japonica* was higher than that of *P. quinquefolia* in the same habitat during the same drought period.

ETR is an expression of the apparent photosynthetic electron transfer rate of plant leaves, which is one of the variables characterizing the photosynthetic capacity of plants and can provide a reference for the evaluation of photosynthetic capacity [43].

As shown in Figure 9, *L. japonica*’s ETR in the L-RO habitat was significantly lower than that in the N-RO habitat during the non- and moderate drought periods; the ETR of *P. quinquefolia* in the L-RO habitat was significantly lower than that in the other two habitats during the mild drought period.

With the development of drought degree, the ETR of *L. japonica* and *P. quinquefolia* significantly declined in all habitats. Overall, the ETRs of *L. japonica* were always greater than those of *P. quinquefolia* in the same habitat during the same drought period.

## 3. Discussion

In this study, the leaf intra- and extracellular water metabolisms of *L. japonica* and *P. quinquefolia* under different simulated RO habitats and synoptic drought conditions were evaluated by determining the electrophysiological and photosynthetic and fluorescence parameters, and the specific water metabolism response mechanisms of these two plants under different stress conditions were summarized as follows:

### 3.1. Analysis of the Intra- and Intercellular Water Metabolic Response Mechanisms of L. japonica Leaves in Heterogeneous RO Habitats

For the three simulated habitats, as the rock-to-soil ratio increased in the habitat, the soil became enriched in bicarbonate (HCO_3_^−^), which was the highest in the L-RO (rock-to-soil ratio: 1) habitat, followed by the S-RO (rock-to-soil ratio: 1/4) habitat and the lowest in the N-RO (rock-to-soil ratio: 0) habitat. Due to the difference in bicarbonate content between the three habitats, the water metabolic response of plant leaves in these habitats also differed. In general, plant leaves close their stomata under bicarbonate stress to avoid excessive water loss, but this behavior also makes it more difficult for the cells to absorb water from the environment, causing the water content of leaves to decrease and the intracellular water status of the leaves to deteriorate, further restricting photosynthetic metabolism; in severe cases, leaf cell structure is damaged and photosynthetic capacity lost, which in turn accelerates the death of the organism.

However, in this paper, we found that the soil in the RO habitats was rich with bicarbonate, the water status of *L. japonica* leaves did not deteriorate but was better than that in the N-RO habitat, and this anomalous phenomenon became more pronounced as the bicarbonate content of the habitat increased (Figure 10). This was attributed to the high CA activity of *L. japonica* [44], which effectively catalyzed the conversion of intracellularly enriched HCO_3_^−^ to water and carbon dioxide, a process that optimized the intracellular water and ionic environment of leaf cells, relieving the excessive demand for extracellular water and ensuring the stomata remained open. However, despite this, the P_N_ of *L. japonica* did not show an increasing trend to match the changes in intracellular water. This implied that the photosynthetic rate of *L. japonica* was mainly limited by non-stomatal factors [45], and high pH stress triggered by the enrichment of bicarbonate in RO habitats could inhibit the synthesis of chlorophyll and Fd, which in turn affected the efficiency of photosynthetic electron transfer and reduced photosynthetic efficiency.

On this basis, this paper further considered the effect of karst drought in karst habitats and investigated the changes in water status of plants under different drought periods in the above heterogeneous simulated RO habitats. Under drought stress, the plants had more difficulty obtaining water and materials from the environment by themselves, which would further amplify the superiority and inferiority of the water metabolism ability of plants under karst adversity. However, combining the results of various parameters, it could be confirmed that although drought stress continued to decrease the water content of the leaves (LWC), the intracellular water-holding capacity (IWHC) and transfer rate (IWTR) were decreasing, water-use efficiency (WUE) was increasing, and the intra- and intercellular water metabolism of the leaves was deteriorating, but in terms of the intracellular water, the plants in RO habitats still exhibited better status than those in the N-RO habitat with the supplement of bicarbonate reverse conversion water. Furthermore, as the drought stress increased, this advantage effectively weakened the degree of water deficit and ionic stress to which the photosynthetic structure of the leaves was subjected, so that the photosynthetic efficacy and its variation of *L. japonica* in RO habitats under drought conditions exhibited no significant difference from that of the control habitat. In other words, *L. japonica* was well adapted to the RO habitats, and this suitability was not altered by the effects of drought.

### 3.2. Analysis of the Intra- and Intercellular Water Metabolic Response Mechanisms of P. quinquefolia Leaves in Heterogeneous RO Habitats

Different from *L. japonica*, *P. quinquefolia* has lower CA activity [42], and its low bicarbonate utilization leads to the excessive intracellular accumulation of bicarbonate in leaves, amplifying the extent to which water metabolism in *P. quinquefolia* is affected by bicarbonate (Figure 10). In the three simulated habitats with different bicarbonate concentrations, the water metabolism of *P. quinquefolia* leaf cells showed a typical trend of deteriorating with increased bicarbonate concentration. Compared to the N-RO habitat, the *P. quinquefolia* in RO habitats had lower gs, and the LWC and intracellular water status IWHC and IWTR decreased with increased bicarbonate concentration in the simulated habitats, weakened photosynthetic capacity and reduced the photosynthetic efficiency.

The water metabolism of *P. quinquefolia* in heterogeneous RO habitats under drought stress was further investigated, and the results showed that the water metabolism process of leaf cells of *P. quinquefolia* in RO habitats were constrained further with the onset and development of drought in the simulated habitats. Comparing leaves of *P. quinquefolia* during different drought periods in the L-RO and N-RO habitats, the LWC, IWHC, IWTR, gs and P_N_ of *P. quinquefolia* in the L-RO habitat decreased significantly during the mild drought period, and gs and P_N_ were even close to zero when the drought further developed to the moderate period; by comparison, although the intra- and intercellular water metabolism capacity of *P. quinquefolia* growing in the N-RO habitat was inhibited, the degree of inhibition was still lower than that in the RO habitats.

## 4. Materials and Methods

### 4.1. Simulation of Heterogeneous RO Habitats in Natural Environment

By adjusting the ratio of rock to soil in indoor pots, the large rock outcrop (L-RO) habitats with outcrops over 1.5 m in height in natural environment are simulated by the rock-to-soil ratio of 1:1 as shown in Figure 11a, the small rock outcrop (S-RO) habitats are simulated by the rock-to-soil ratio of 1:4 (Figure 11b), and the no rock outcrop (N-RO) habitats are taken as control. Each simulated habitat was regularly sprayed with equal amounts of tap water (1 L per 3 days/time) and lasted for two months.

### 4.2. Plant Material and Water Control

The experimental plants were selected from annual “Beihua No. 1” *L. japonica* and *P. quinquefolia* seedlings, which were moved into three simulated habitats in April 2022 and observed in July 2022. Culture environment: photosynthetic photon flux density (PPFD) in the lab was 500 µmol·m^−2^·s^−1^, daily light time 12 h, indoor temperature maximum 25 °C, minimum 16 °C, indoor air humidity 65–75%.

Three seedlings (3 pots) of honeysuckle and *P. quinquefolia* growing robustly in three simulated habitats were selected for the measurement. Different levels of drought stress were simulated using artificially controlled moisture. Each simulated habitat was supplemented with water 2 days prior to the experimental observation so that its habitat water reached saturation and a 10-day drought treatment was started after 2 days of stabilization. Soil moisture data were obtained with the aid of HydraGO probes (seeing Table 4) for each simulated habitat from 7:00 a.m. to 8:00 a.m. According to the change in soil water content, the photosynthetic, fluorescence and electrophysiological parameters of the experimental plants were measured when the soil water content was about 25% (non-drought), 15% (mild drought) and 8% (moderate drought) [46].

### 4.3. Theory and Methods for Obtaining Index Parameters

#### 4.3.1. Selection of Plant Leaf Materials and Data Collection Time

The fully expanded leaves at the 4th or 5th leaf position of the newborn branches of *L. japonica* and *P. quinquefolia* were selected for data collection. The data collection time was from 8:00 to 10:00 on the same day.

#### 4.3.2. The Obtaining of Photosynthetic Parameters

The photosynthetic and fluorescence indices of the plants were measured using a portable photosynthetic measurement system Li-6800 (Li-Cor Inc., Lincoln, NE, USA), and the measurement was conducted in triplicate for each plant species.

Photosynthesis was measured using red and blue light source leaf chamber and a CO_2_ cylinder with a PPFD of 500 µmol·m^−2^·s^−1^, temperature of 25 °C and CO_2_ concentration of 400 µmol·mol^−1^.

Photosynthetic indicators: net photosynthetic rate (P_N_, µmol·m^−2^·s^−1^), stomatal conductance (gs, mol·m^−2^·s^−1^) and transpiration rate (E, mmol·m^−2^·s^−1^) of plant leaves. The instantaneous water-use efficiency (WUE_i_, µmol·mmol^−1^) was:(1)$${\mathrm{WUE}}_{i}{\;=P}_{N}/E$$

#### 4.3.3. The Obtaining of Fluorescence Parameters

The chlorophyll fluorescence was measured with a portable photosynthetic measurement system Li-6800. Plants were fully dark-adapted for 30 min to ensure that the PSII reaction center of the plant leaves was completely opened before measurement. The fluorescence parameters of original fluorescence (F_o_), maximum fluorescence (F_m_), photochemical quenching coefficient (qP) and electron transfer rate (ETR) of leaves were measured and record, and the maximum photochemical efficiency of PSⅡ (F_v_/F_m_) was calculated as follows:(2)$$F_{v}{/F}_{m}{\;=(F}_{m}-F_{o}{)/F}_{m}$$

#### 4.3.4. Calculation of the Water Content of Plant Leaves

The water content of the leaves (LWC, %) was measured using the international drying constant weight method. The fresh leaves were numbered and weighed on an analytical balance and recorded as W_f_, and then the leaves were put into wide-mouth bottles and the bottles were put into an oven at 80 °C for 10–12 h to dry to constant weight, which was weighed and recorded as W_d_. The LWC was calculated as follows:(3)$$\mathrm{LWC}=\frac{W_{f}-W_{d}}{W_{f}}\times 100$$

#### 4.3.5. The Theory and Modeling of Electrophysiological Parameters

As electrophysiological parameters are easily influenced by environmental stimuli, the isolated leaves were firstly immersed in water for 30 min to ensure that the leaf cells were in a standard and uniform state. Afterwards, the surface of each leaf was wiped off with a paper towel and sandwiched between parallel electrode plates as shown in Figure 12 [31] and then connected to an LCR tester (model 6300, GWinstek Electronics Industrial Co., Ltd., Taiwan, China). Because the electrical characteristics of plant leaves are low capacitance and high impedance, the measurement frequency and voltage were set to 3 KHz and 1.5 V and employed parallel mode. The physiological impedance (Z), physiological capacitance (C) and physiological resistance (R) were measured by sequentially adding weights of equal mass (M = 100 g) to the leaves with different gradients of 1.17 N, 2.17 N, 3.17 N, 5.17 N and 7.17 N of clamping force (F, N). Three different parts of each leaf were measured, the average value was taken, and the measurements were repeated three times at each level for each treatment. Previous studies by our group showed that the plant leaves can withstand forces ranging from 15.89 to 30.01 N. Therefore, the effect of the maximum clamping force of 7.17 N on the measurement results in this experiment is negligible [47].

The calculated equation for intracellular water-holding capacity (IWHC), water transport rate (IWTR) and water-use efficiency (IWUE) are as follows [45]:(4)$$\left\{\begin{array}{l} \mathrm{IWHC}=\sqrt{{\mathrm{IC}}^{3}}\\ \mathrm{IWTR}=\frac{\mathrm{IWHC}}{\mathrm{IC}\times \mathrm{IZ}}\\ \mathrm{IWUE}=\frac{d}{\mathrm{IWHC}} \end{array}\right.$$

### 4.4. Data Statistics and Analysis

Valid data were screened and obtained by Microsoft Excel 2019. The electrophysiological parameter curves were fitted using SigmaPlot 14.0 software. Analysis of variance was performed using DPS software, multiple comparisons were performed using the Tukey method, and the level of significant differences was assessed using the lowest significant difference post hoc test at the 5% level of significance (*p* ≤ 0.05). Data are shown as mean ± standard error determined by one-sample *t*-test. Graphs were plotted using Origin 2019.

## 5. Conclusions

This study revealed the differential water metabolic responses of *L. japonica* and *P. quinquefolia* to heterogeneous RO habitats based on analysis of the water content of plant leaves and leaf electrical and photosynthetic properties. It was found that the water metabolism of *P. quinquefolia* was significantly inhibited under karst bicarbonate enrichment, and this inhibition would be further amplified by the occurrence of karstic drought, making it difficult for it to adapt to such habitats. Compared with the *P. quinquefolia*, the water metabolism mechanism of *L. japonica* was the basis for its ability to be adapted to karst habitats. *L. japonica* could efficiently use its internal enriched bicarbonate, and this resulted in a better water status of *L. japonica* growing in the RO habitats with a high concentration of bicarbonate, its stronger intracellular water-holding capacity effectively regulating the leaf intra- and intercellular water metabolic environment, which was probably the reason for the high photosynthetic capacity of the plant in the dry habitat of the rock outcrops, and the stable intracellular water-use efficiency also made it more adapted to the karst drought environment. The results in this research deepen the understanding of the response mechanisms of plant water metabolism to heterogeneous karst habitats to a certain extent and can provide a useful reference for the screening of pioneer species in karst adversity restoration.

## Figures and Tables

**Figure 1 plants-12-02279-f001:**
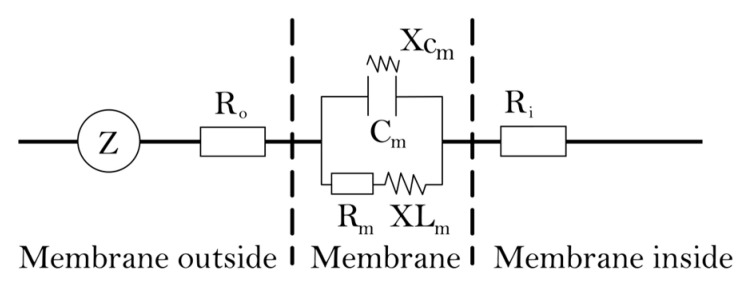
Leaf cell equivalent circuit diagram. Note: Z is impedance, C_m_ is film capacitance, R_m_ is film resistance, Xc_m_ is film capacitance, XL_m_ is film inductance, R_o_ is external film resistance, R_i_ is internal film resistance.

**Figure 2 plants-12-02279-f002:**
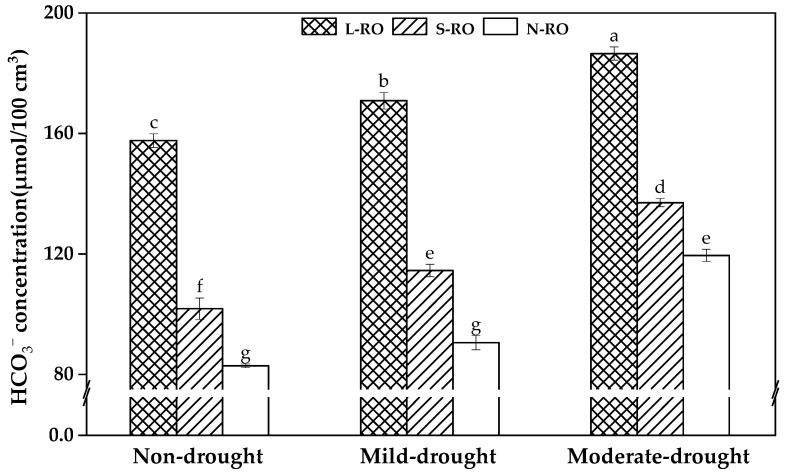
Concentrations of HCO_3_^−^ in soils of simulated habitats under non-drought, mild and moderate drought conditions. Note: data in the figure are mean ± standard deviation, n = 3. The different lowercase letters indicate significant differences at 5 level *p* < 0.05.

**Figure 3 plants-12-02279-f003:**
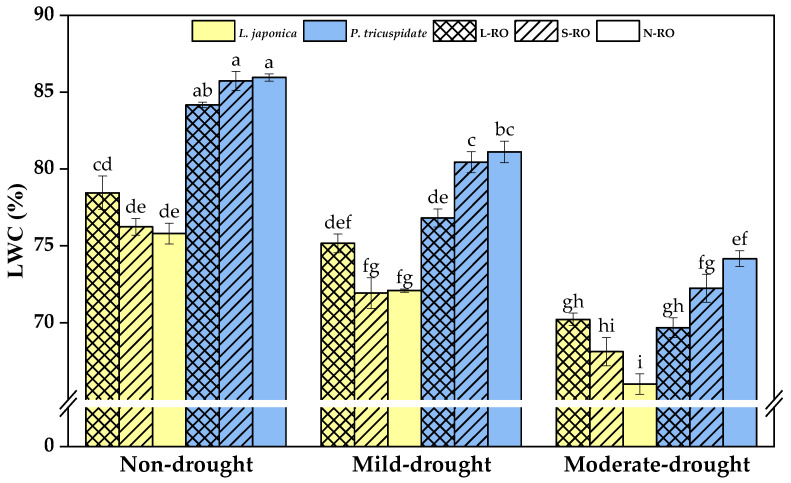
Leaf moisture content (LWC) of *L. japonica* and *P. quinquefolia* during non-drought, mild drought and moderate drought periods in different simulated habitats. The different lowercase letters indicate significant differences at 5 level *p* < 0.05.

**Figure 4 plants-12-02279-f004:**
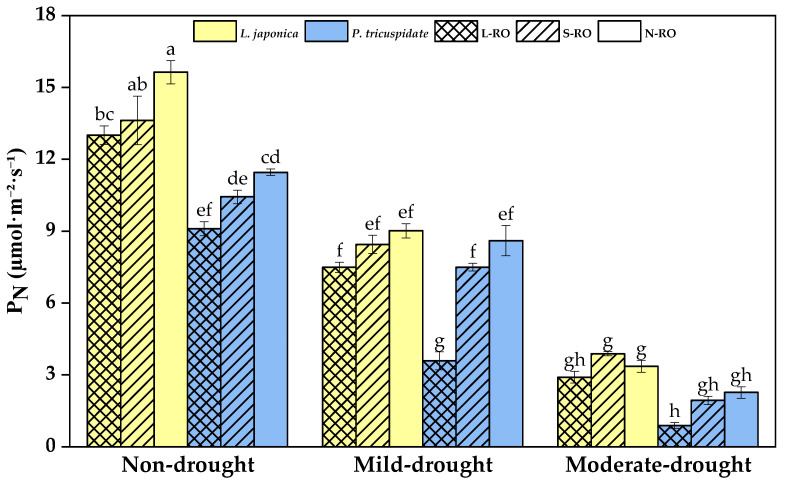
Net photosynthetic rate (P_N_) of *L. japonica* and *P. quinquefolia* during non-drought, mild drought and moderate drought periods in different simulated habitats. Note: Data in the figure are mean ± standard deviation, n = 3, and different lowercase letters indicate significant differences (*p* < 0.05), which is the same below.

**Figure 5 plants-12-02279-f005:**
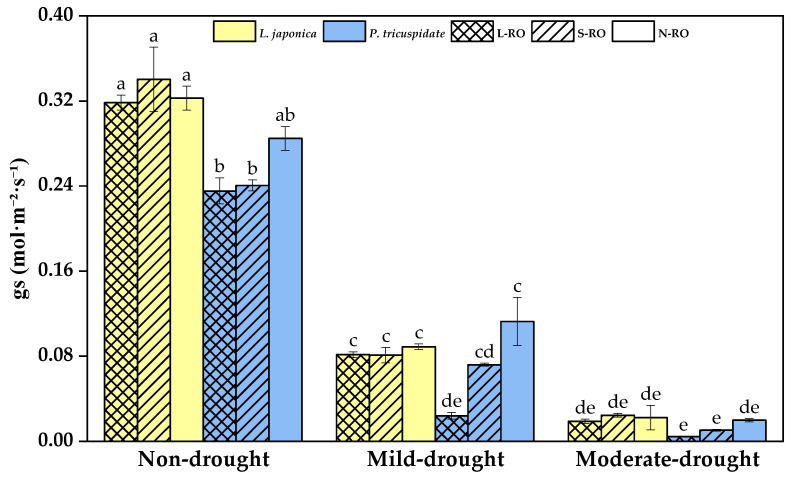
Stomatal conductance (gs) of *L. japonica* and *P. quinquefolia* during non-drought, mild drought and moderate drought periods in different simulated habitats. The different lowercase letters indicate significant differences at 5 level *p* < 0.05.

**Figure 6 plants-12-02279-f006:**
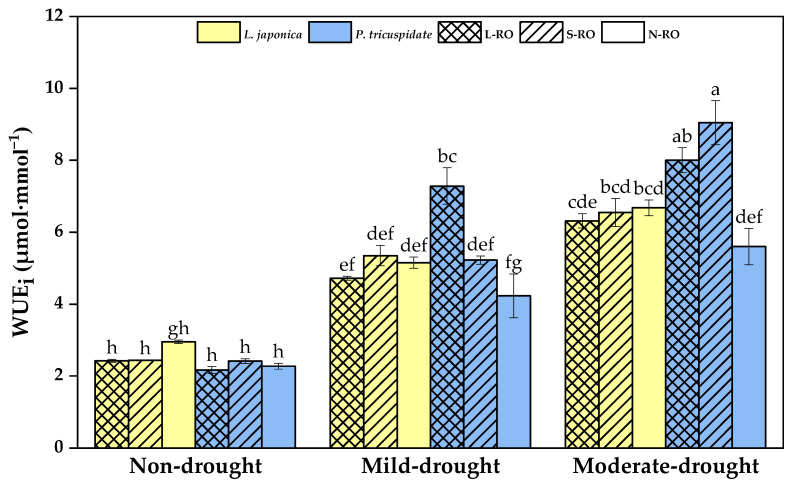
Instantaneous water-use efficiency (WUE_i_) of *L. japonica* and *P. quinquefolia* during non-drought, mild drought and moderate drought periods in different simulated habitats. The different lowercase letters indicate significant differences at 5 level *p* < 0.05.

**Figure 7 plants-12-02279-f007:**
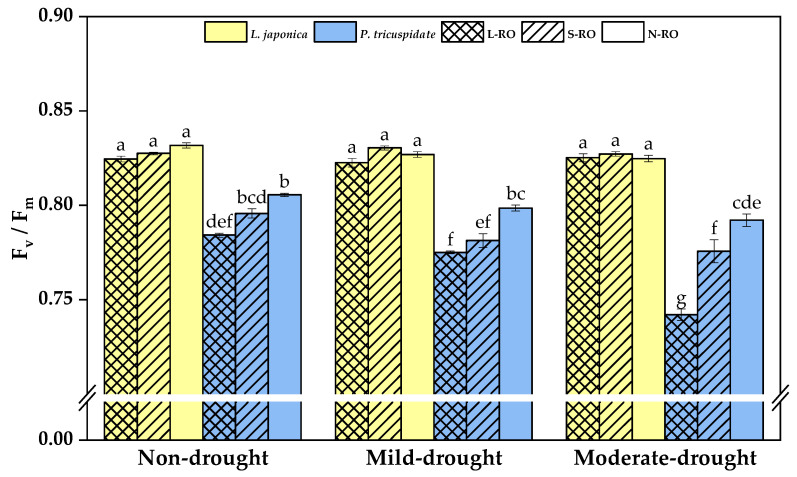
F_v_/F_m_ of *L. japonica* and *P. quinquefolia* during non-drought, mild drought and moderate drought periods in different simulated habitats. The different lowercase letters indicate significant differences at 5 level *p* < 0.05.

**Figure 8 plants-12-02279-f008:**
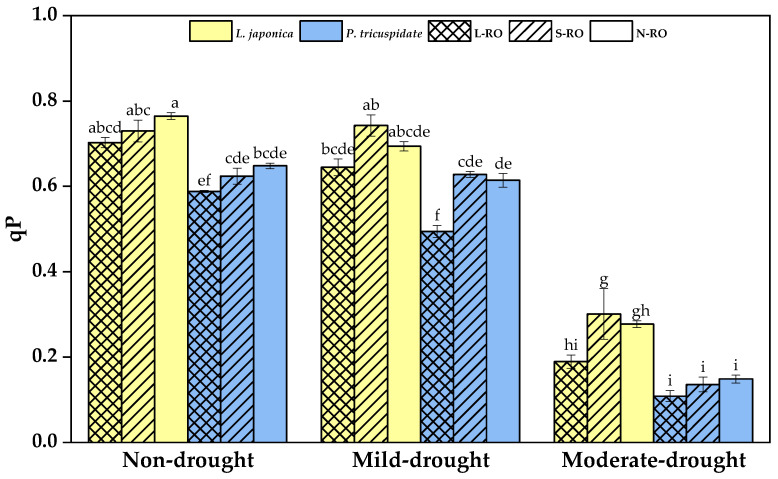
Photochemical fluorescence (qP) of *L. japonica* and *P. quinquefolia* in different simulated habitats during non-drought, mild drought and moderate drought periods. The different lowercase letters indicate significant differences at 5 level *p* < 0.05.

**Figure 9 plants-12-02279-f009:**
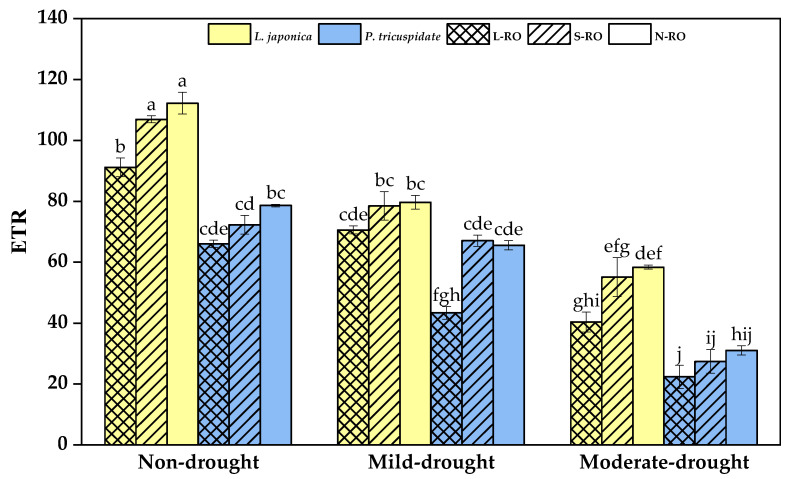
Photosynthetic electron transfer rate (ETR) of *L. japonica* and *P. quinquefolia* during non-drought, mild drought and moderate drought periods in different simulated habitats. The different lowercase letters indicate significant differences at 5 level *p* < 0.05.

**Figure 10 plants-12-02279-f010:**
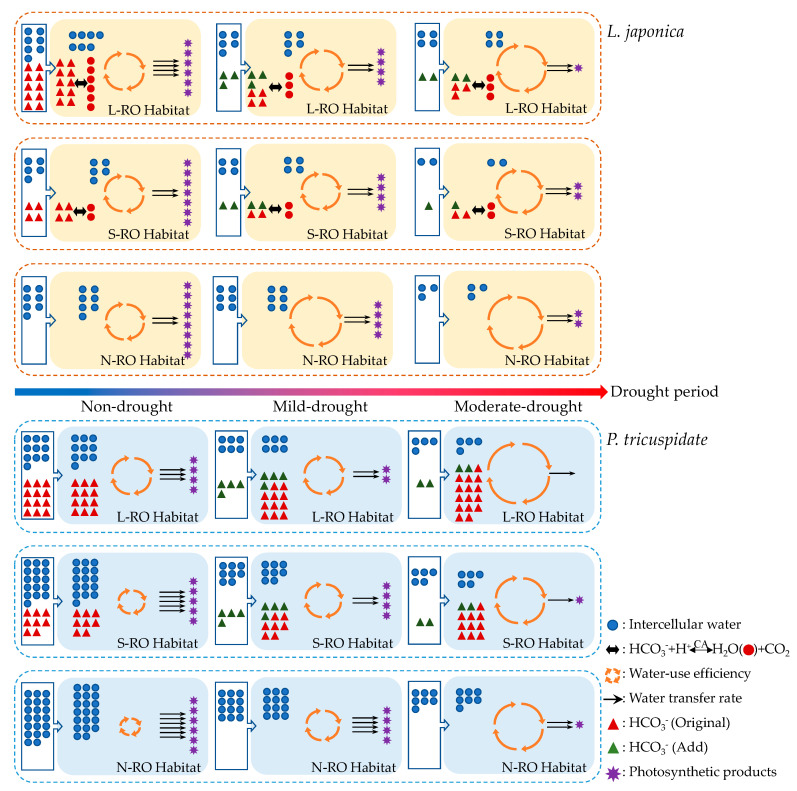
Dynamic traits of leaf water metabolism of *L. japonica* and *P. quinquefolia* in three simulated habitats during different drought periods.

**Figure 11 plants-12-02279-f011:**
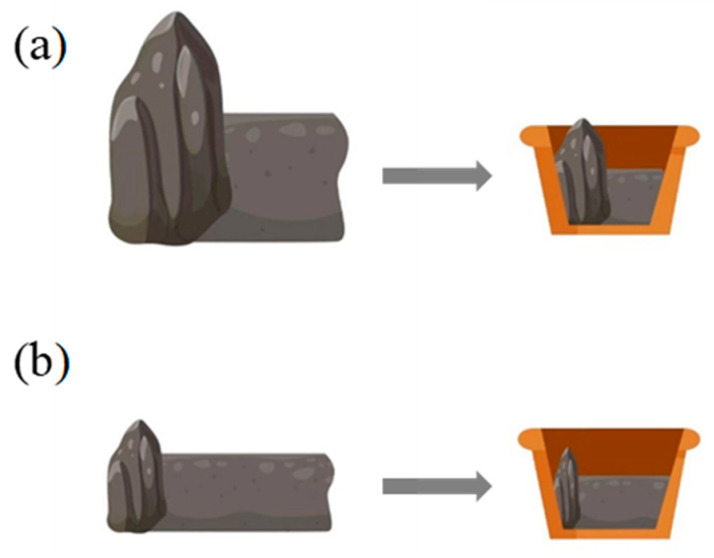
Schematic diagram of simulated RO habitats: (**a**) L-RO habitats; (**b**) S-RO habitats.

**Figure 12 plants-12-02279-f012:**
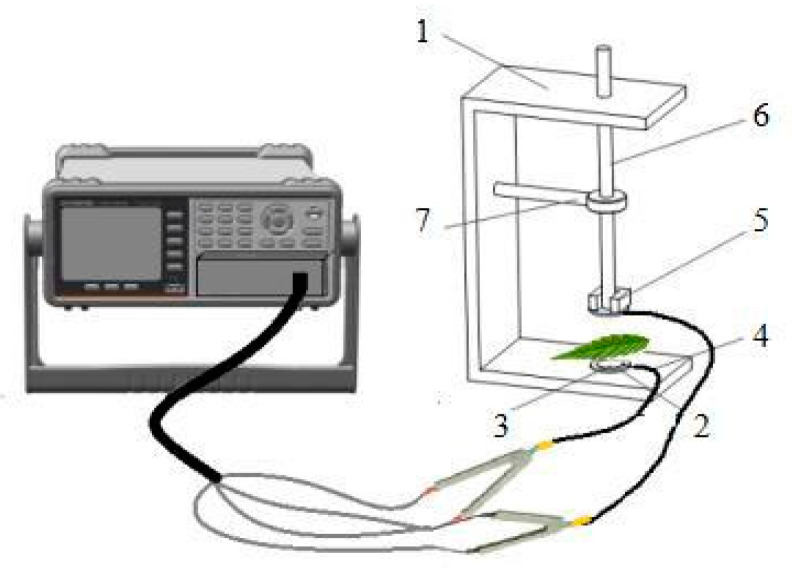
Schematic diagram of parallel plate electrode and experimental test setup [31]: (1) bracket; (2) foam plate; (3) electrode plate; (4) electric wire; (5) iron block; (6) plastic rod; (7) fixing clip.

**Table 1 plants-12-02279-t001:** Intracellular water-holding capacity (IWHC) of leaves of *L. japonica* and *P. quinquefolia* under different drought periods in three simulated habitats.

Plant Species	Drought Period	Simulated Habitat		
		L-RO	S-RO	N-RO
*L. japonica*	Non-drought	388.73 ± 21.33 b	207.51 ± 18.82 cdef	206.70 ± 13.32 cdef
	Mild	256.81 ± 31.15 cd	178.06 ± 25.20 defg	181.18 ± 1.62 defg
	Moderate	204.25 ± 18.82 cdef	122.48 ± 16.91 efg	92.20 ± 6.32 g
*P. quinquefolia*	Non-drought	296.87 ± 9.05 de	502.43 ± 18.42 a	599.11 ± 43.01 a
	Mild	176.80 ± 4.08 defg	243.16 ± 23.80 cd	376.03 ± 22.84 b
	Moderate	113.90 ± 5.48 fg	153.80 ± 19.74 defg	221.39 ± 15.49 cde

Note: values indicate the mean ± SD, n = 3, and different lowercase letters differ significantly (*p* < 0.05) in the content between habitats during different drought periods.

**Table 2 plants-12-02279-t002:** Intracellular water transfer rate (IWTR) of *L. japonica* and *P. quinquefolia* leaves under different drought periods in three simulated habitats.

Plant Species	Drought Period	Simulated Habitat		
		L-RO	S-RO	N-RO
*L. japonica*	Non-drought	7.92 ± 0.85 bc	4.07 ± 0.37 efgh	4.04 ± 0.26 efgh
	Mild	4.99 ± 0.55 ef	3.50 ± 0.48 efghi	3.48 ± 0.04 efghi
	Moderate	4.02 ± 0.21 efgh	2.50 ± 0.40 ghi	1.77 ± 0.12 i
*P. quinquefolia*	Non-drought	5.60 ± 0.25 de	9.64 ± 0.80 ab	11.70 ± 0.78 a
	Mild	3.33 ± 0.06 fghi	4.64 ± 0.44 efg	7.21 ± 0.43 cd
	Moderate	2.16 ± 0.10 hi	2.92 ± 0.37 fghi	4.23 ± 0.29 efgh

Note: values indicate the mean ± SD, and different lowercase letters indicate significant differences at 5 level *p* < 0.05.

**Table 3 plants-12-02279-t003:** Intracellular water-use efficiency (IWUE) of *L. japonica* and *P. quinquefolia* under different drought periods in three simulated habitats.

Plant Species	Drought Period	Simulated Habitat		
		L-RO	S-RO	N-RO
*L. japonica*	Non-drought	0.12 ± 0.00 bcd	0.12 ± 0.01 bcde	0.11 ± 0.01 cde
	Mild	0.13 ± 0.01 bcd	0.12 ± 0.00 bcde	0.16 ± 0.00 abc
	Moderate	0.15 ± 0.01 abc	0.14 ± 0.01 abc	0.17 ± 0.01 ab
*P. quinquefolia*	Non-drought	0.12 ± 0.01 bcde	0.08 ± 0.01 de	0.06 ± 0.00 e
	Mild	0.12 ± 0.01 bcde	0.12 ± 0.01 bcde	0.11 ± 0.01 cde
	Moderate	0.19 ± 0.03 a	0.15 ± 0.02 abc	0.15 ± 0.02 abc

Note: values indicate the mean ± SD, and different lowercase letters indicate significant differences at 5 level *p* < 0.05.

**Table 4 plants-12-02279-t004:** Soil moisture in the experimental simulated habitats under different drought days.

Plant Species	Simulated Habitat	Soil Moisture (%)									
		Day 1	Day 2	Day 3	Day 4	Day 5	Day 6	Day 7	Day 8	Day 9	Day 10
*L japonica*	L-RO	33.53 ± 0.27	27.57 ± 0.32	23.93 ± 0.03	20.03 ± 0.12	15.87 ± 0.54	12.73 ± 0.30	10.67 ± 0.41	9.73 ± 0.18	7.50 ± 0.21	6.10 ± 0.21
	S-RO	35.73 ± 0.19	31.33 ± 0.24	26.20 ± 0.12	21.53 ± 0.19	17.03 ± 0.15	14.50 ± 0.32	11.30 ± 0.36	9.33 ± 0.21	7.87 ± 0.24	6.90 ± 0.06
	N-RO	35.53 ± 0.12	31.70 ± 0.15	26.77 ± 0.18	24.83 ± 0.63	18.53 ± 0.32	14.23 ± 0.37	10.63 ± 0.32	9.03 ± 0.18	7.97 ± 0.09	7.00 ± 0.12
*P. quinquefolia*	L-RO	32.80 ± 0.40	27.07 ± 0.12	20.87 ± 0.13	14.57 ± 0.43	13.53 ± 0.33	12.00 ± 0.36	9.80 ± 0.43	8.43 ± 0.27	7.87 ± 0.21	6.03 ± 0.15
	S-RO	34.70 ± 0.21	31.63 ± 0.26	26.93 ± 0.07	18.77 ± 0.12	14.97 ± 0.18	13.73 ± 0.64	11.87 ± 0.27	8.37 ± 0.27	7.60 ± 0.23	6.57 ± 0.22
	N-RO	34.03 ± 0.12	30.77 ± 0.19	26.1 ± 0.12	19.70 ± 0.21	16.03 ± 0.38	13.30 ± 0.40	11.37 ± 0.22	9.50 ± 0.35	8.10 ± 0.42	6.70 ± 0.35

Note: data in the table are mean ± standard deviation, water content during non-drought, mild drought and moderate drought periods are marked in blue, orange and red, respectively.

## Data Availability

The data presented in this study are available on request from the corresponding author.

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
