# Peer review of "Water Metabolism of Lonicera japonica and Parthenocissus quinquefolia in Response to Heterogeneous Simulated Rock Outcrop Habitats"

_plants, 2023, doi:10.3390/plants12122279_

Round 1
Reviewer 1 Report
Dear authors
The manuscript entitled (Water metabolism of Lonicera japonica and Parthenocissus quinquefolia in response to heterogeneous simulated rock outcrops habitats) is well presented with novel idea on electrophysiological changes of Lonicera japonica and Parthenocissus quinquefolia , but very minor comments were included:
Arrange the keywords in alphabetic order.
Arrange the keywords in alphabetic order.
Figure 1 need to be in high resolution.
Line 95, write a brief information about electrophysiological parameters.
The rationale behind the work needs to be more highlighted.
Change one or two tables to figures.
Improve the aim of the work.
Provide a figure describing the merit of your study.
Author Response
- Comment: Arrange the keywords in alphabetic order.
Reply: Thanks for your comment. According to the reviewers' suggestions, we have rearranged the keywords in alphabetical order.
- Comment:Figure 1 need to be in high resolution.
Reply: Thanks for your comment. We have changed Figure 1 to a higher resolution image.
- Comment: Line 95, write a brief information about electrophysiological parameters.
Reply: Thanks for your good suggestion, we have added some information about the electrophysiological parameters into the corresponding part, as shown in lines 100-104 in the revised manuscript.
- Comment:The rationale behind the work needs to be more highlighted.
Reply: Thanks for your comment. In the revised manuscript we have added a description of the significance of this study (Line:53-58).
- Comment:Change one or two tables to figures.
Reply: Thanks for your suggestion. We have converted Table1 into a Figure, which is shown in Figure 2 of the manuscript (Line:165).
- Comment:Improve the aim of the work.
Reply: Thanks for your good comment, we have revised the last paragraph of the Introduction part to improve the aim of the work (Line:145-154).
- Comment:Provide a figure describing the merit of your study.
Reply: Thanks to your comments, we have added a graphic abstract.
Reviewer 2 Report
The manuscript of Zhao et al. "Water metabolism of Lonicera japonica and Parthenocissus quinquefolia in response to heterogeneous simulated rock out crops habitats" is devoted to the study of the influence of HCO3- content in the soil on the activity of the most important physiological processes - photosynthesis and transpiration. The topic of the work is relevant. Plant habitats vary greatly in their characteristics. Such specific conditions as rock outcrops are little explored. The particular relevance of the study is related to the study of the effects of drought on plants that grow in such specific conditions. The authors performed studies on two plant species - Lonicera japonica and Parthenocissus quinquefolia. The study showed that P. quinquefolia plants are more affected by drought than L. japonica plants. For P. Quinquefolia plants, the negative effect of high HCO3- content is more pronounced.
There are a number of questions.
First of all, it should be noted that the method of recording the water status of plants used in the work is indirect. The calculation of the indices is based on the measurement of the impedance. However, the calculation of indices is based on a number of assumptions. The use of such indices requires verification.
The authors make a conclusion about the change in the water status of plants. However, the manuscript lacks information about the water content of the plants. Results of measurement of fresh weight of leaves, dry weight of leaves, relative water content in leaves are necessary. Only in this case it is possible to speak about a change in the water status.
Results of leaf water content combined with soil moisture measurements (demonstrated) and stomatal conductance (demonstrated) will allow assessment of changes in plant water balance.
Before measuring the impedance, the leaf was immersed in water for 30 minutes. During this time, the amount of water in the leaf will change significantly. Even indirect measurement of water status will be distorted.
The Materials and Methods section provides a detailed description of the calculation of indices. The above description repeats the previous work of the authors Zhang, C.; Wu, Y.Y.; Su, Y.; Li, H.T.; Fang, L.; Xing, D.K. (2021). Plant's electrophysiological information manifests the composition and nutrient transport characteristics of membrane proteins. Plant Signaling & Behavior, 2021, 16(7), 1918867. In the manuscript, it is sufficient to cite the published work.
In the Discussion section, it is suggested that carbonic anhydrase activity can compensate for the lack of water. An improvement in water status is expected due to the release of water as a result of the splitting of HCO3-. What is the mass fraction of HCO3- in the solution absorbed by the plant from the soil? Judging by the data given in Table 1, it is about 0.001%. And the water content in the soil during drought changes by tens of%. What kind of compensation can we talk about?
In the introduction, it is better to make references to articles where the passive electrical properties of plants are studied. Quite a lot of work has been devoted to the study of the passive electrical properties of plants.
The manuscript abstract does not contain specific information about reported effects.
Author Response
- Comment: First of all, it should be noted that the method of recording the water status of plants used in the work is indirect. The calculation of the indices is based on the measurement of the impedance. However, the calculation of indices is based on a number of assumptions. The use of such indices requires verification.
Reply: Thanks for your comments. We acknowledge that the electrophysiological parameters in this study were obtained indirectly based on measurements of leaf physiological capacitance(C), resistance(R) and impedance(Z), but they have also been widely used in various studies because of their inescapable advantage in reflecting the water status of plant leaves, especially intracellular water status. For example, Jamaludin et al. (2015) used impedance measurements to accurately reflect the internal water status of plants; Yu et al. (2021) showed that the electrophysiological information C, R and Z of plant leaves could better characterize the response of plant growth and development to soil moisture; Zhang et al. (2020) used electrophysiological parameters IWHC, IWUE and IWHT to rapidly monitor the physiological status of plants, and the intracellular water metabolism strategies of plants in response to environmental changes were revealed; Qin et al. (2022) used electrophysiological parameters IWTR and IWUE to analyze the strategies of different adapted plants in response to the Karst drought environment. Thus, we chose electrophysiological parameters to characterize plant intracellular water metabolism in this study as well. For clarification, we have added some sentences to discuss that in the revised manuscript. (Line 104-110)
Jamaludin D, Abd Aziz S, Ahmad D, et al. Impedance analysis of Labisia pumila plant water status[J]. Information Processing in Agriculture, 2015, 2(3-4): 161-168.
Yu R, Wu Y, Xing D. Can electrophysiological parameters substitute for growth, and photosynthetic parameters to characterize the response of mulberry and paper mulberry to drought?[J]. Plants, 2021, 10(9): 1772.
Zhang C, Wu Y, Su Y, et al. A plant’s electrical parameters indicate its physiological state: a study of intracellular water metabolism[J]. Plants, 2020, 9(10): 1256.
Qin X, Xing D, Wu Y, et al. Diurnal Variation in Transport and Use of Intracellular Leaf Water and Related Photosynthesis in Three Karst Plants[J]. Agronomy, 2022, 12(11): 2758.
- Comment: The authors make a conclusion about the change in the water status of plants. However, the manuscript lacks information about the water content of the plants. Results of measurement of fresh weight of leaves, dry weight of leaves, relative water content in leaves are necessary. Only in this case it is possible to speak about a change in the water status.
Reply: Thanks for your comment. We have added the water content of plant leaves information into the revised manuscript (Line:175-190,443-449).
- Comment: Results of leaf water content combined with soil moisture measurements (demonstrated) and stomatal conductance (demonstrated) will allow assessment of changes in plant water balance.
Reply: Thanks for your comment. The combination of leaf water content with soil moisture and stomatal conductance is an undeniable method for assessing the plant water balance, which helps to evaluate the adaptability of plant. But in fact, the dynamics of the leaf intracellular water were the major concern of this study, as the use of water retained within leaf cells is critical for their growth and development, through the investigation of the dynamics of the leaf intracellular water, we intended to explore the water metabolic characteristics of plants in heterogeneous RO habitats under synoptic drought conditions. Furthermore, the electrophysiological indices used to reflect the intracellular water status in this study can be easily and rapidly determined, the electrophysiological technique can help improve the efficiency in screening the adaptive plants.
- Comment: Before measuring the impedance, the leaf was immersed in water for 30 minutes. During this time, the amount of water in the leaf will change significantly. Even indirect measurement of water status will be distorted
Reply: Thanks for your comment. The purpose of the 30-minute immersion was to ensure that all the plant leaf cells in each treatment reached a standard state at that moment, by doing this, the different of the dynamics of the leaf intracellular water at each treatment was mainly caused by the RO habitats and drought level, and they could be better compared. As such, the information obtained from the electrophysiological parameters could purposefully reflect the metabolic characteristics of the water within leaf cells at that moment, which helped to analyze the relationship between the RO habitats and drought and plant leaf intracellular water. We intended to compare the dynamics of the leaf intracellular water, which is related to the permeability of the cell membrane. The leaf water status could then be better investigated by analyzing the variation of leaf water content together with the dynamics of the leaf intracellular water.
- Comment:The Materials and Methods section provides a detailed description of the calculation of indices. The above description repeats the previous work of the authors Zhang, C.; Wu, Y.Y.; Su, Y.; Li, H.T.; Fang, L.; Xing, D.K. (2021). Plant's electrophysiological information manifests the composition and nutrient transport characteristics of membrane proteins. Plant Signaling & Behavior, 2021, 16(7), 1918867. In the manuscript, it is sufficient to cite the published work.
Reply: Thanks for your kind suggestion, we have simplified the description of the calculation of the electrophysiological parameters in the Materials and Methods section in the revised manuscript.
- Comment:In the Discussion section, it is suggested that carbonic anhydrase activity can compensate for the lack of water. An improvement in water status is expected due to the release of water as a result of the splitting of HCO3-. What is the mass fraction of HCO3- in the solution absorbed by the plant from the soil? Judging by the data given in Table 1, it is about 0.001%. And the water content in the soil during drought changes by tens of%. What kind of compensation can we talk about?
Reply: Thanks for your comment. This study mainly aimed to explore the water metabolism characteristics of potentially suitable karst plants in a high bicarbonate concentration and arid karst habitat, the H2O converted from HCO3- which was catalyzed by CA could change the intracellular water status, and had influence on the variation of corresponding electrophysiological parameters, the detailed compensation quantity was not the objective of this study. But of course, we will conduct further experiments to proceed the problem solving, since the compensation is important for the water requirement investigation. According to our previous studies, the plant water status can be improved by the compensation of water from the splitting of HCO3-, which was catalyzed by the carbonic anhydrase. For example, greater CA activity in B. papyrifera compared with that in M. alba, led to greater bicarbonate use capacity. B. papyrifera had approximately 30% of its photosynthate derived from the CO2 converted by CA under the 10mM NaHCO3 treatment, whereas M. alba had only 0-15%. At this moment, there was also about 30% H2O devoted to the photosynthate in B. papyrifera under the bicarbonate treatment (Wu Y Y, Xing D K. Effect of bicarbonate treatment on photosynthetic assimilation of inorganic carbon in two plant species of Moraceae. Photosynthetica, 2012, 50(4):587-594). The compensation was determined by the CA activity and plant species. But in this study, the compensation ratio in L. japonica and P. quinquefolia might be less than B. papyrifera. However, the conversion of HCO3- to H2O catalyzed by CA could change the intracellular water status, and this phenomenon indeed had effect on the intracellular water metabolism.
- Comment:In the introduction, it is better to make references to articles where the passive electrical properties of plants are studied. Quite a lot of work has been devoted to the study of the passive electrical properties of plants.
Reply: Thanks for your comment. We have added some of the findings on plant electrophysiology into the revised manuscript, specifically in lines 104-110 in the revised manuscript.
- Comment:The manuscript abstract does not contain specific information about reported effects.
Reply: Thanks for your comment. We have modified the abstract to add some parameter information. (Line: 19-33)